# Enablers and Barriers to Accessing Healthcare Services for Aboriginal People in New South Wales, Australia

**DOI:** 10.3390/ijerph18063014

**Published:** 2021-03-15

**Authors:** Davida Nolan-Isles, Rona Macniven, Kate Hunter, Josephine Gwynn, Michelle Lincoln, Rachael Moir, Yvonne Dimitropoulos, Donna Taylor, Tim Agius, Heather Finlayson, Robyn Martin, Katrina Ward, Susannah Tobin, Kylie Gwynne

**Affiliations:** 1The Poche Centre for Indigenous Health, Faculty of Medicine and Health, Edward Ford Building A27, The University of Sydney, Camperdown, NSW 2006, Australia; dnol9371@uni.sydney.edu.au (D.N.-I.); josephine.gwynn@sydney.edu.au (J.G.); rachael.moir@sydney.edu.au (R.M.); yvonne.dimitropoulos@sydney.edu.au (Y.D.); susannahbrodie@gmail.com (S.T.); kylie.gwynne@mq.edu.au (K.G.); 2School of Population Health, Faculty of Medicine and Health, University of New South Wales, Sydney, NSW 2052, Australia; 3Faculty of Medicine, Health and Human Sciences, Macquarie University, Sydney, NSW 2109, Australia; 4Faculty of Medicine and Health, The George Institute for Global Health, University of New South Wales, Newtown, NSW 2042, Australia; khunter@georgeinstitute.org.au; 5Faculty of Medicine and Health, Sydney School of Health Sciences, The University of Sydney, Camperdown, NSW 2006, Australia; 6Faculty of Health, The University of Canberra, Bruce, ACT 2617, Australia; michelle.lincoln@canberra.edu.au; 7Pius X Aboriginal Health Service, Moree, NSW 2400, Australia; ceo@piusx.com.au; 8Durri Aboriginal Corporation Medical Service, 15-19 York Lane, Kempsey, NSW 2440, Australia; tim.agius@y7mail.com; 9Brewarrina Multipurpose Health Service, 56 Doyle Street, Brewarrina, NSW 2839, Australia; Heather.Finlayson@health.nsw.gov.au; 10Mid North Coast Local Health District, Port Macquarie, NSW 2444, Australia; Robyn.Martin3@health.nsw.gov.au; 11Brewarrina Aboriginal Health Service, 5-7 Sandon Street, Brewarrina, NSW 2839, Australia; katrinaw@brewarrinaams.com.au

**Keywords:** indigenous, first nations, health services accessibility, health services administration, trust, communication, primary health care, health policy

## Abstract

Background: Australia’s healthcare system is complex and fragmented which can create challenges in healthcare, particularly in rural and remote areas. Aboriginal people experience inequalities in healthcare treatment and outcomes. This study aimed to investigate barriers and enablers to accessing healthcare services for Aboriginal people living in regional and remote Australia. Methods: Semi-structured interviews were conducted with healthcare delivery staff and stakeholders recruited through snowball sampling. Three communities were selected for their high proportion of Aboriginal people and diverse regional and remote locations. Thematic analysis identified barriers and enablers. Results: Thirty-one interviews were conducted in the three communities (*n* = 5 coastal, *n* = 13 remote, and *n* = 13 border) and six themes identified: (1) Improved coordination of healthcare services; (2) Better communication between services and patients; (3) Trust in services and cultural safety; (4) Importance of prioritizing health services by Aboriginal people; (5) Importance of reliable, affordable and sustainable services; (6) Distance and transport availability. These themes were often present as both barriers and enablers to healthcare access for Aboriginal people. They were also present across the healthcare system and within all three communities. Conclusions: This study describes a pathway to better healthcare outcomes for Aboriginal Australians by providing insights into ways to improve access.

## 1. Introduction

Australia’s healthcare system is structured across three levels of government (Federal, state/territory and local) and eight state/territory jurisdictions with multiple government departments and private and not-for-profit service providers. The federal government deals primarily with resource allocation and national policy, while states and territories primarily manage the delivery of healthcare. All levels of government are engaged in regulation and compliance. [1]. In addition, various healthcare services exist specifically for priority population groups including Aboriginal and Torres Strait Islander peoples and are delivered by non-government organizations and supported by state and federal government funding. These include Aboriginal Community Controlled Health Organizations (ACCHOs) which provide a range of culturally safe primary health care services specifically for local Aboriginal communities [2]. ACCHOs have demonstrated better performance and outcomes than mainstream general practice [2,3]. This research took place in the state of New South Wales (NSW) on unceded Aboriginal land and the term Aboriginal will be used hereafter.

The governance, funding and delivery of healthcare has created a complex and fragmented healthcare system which struggles to deliver effective patient-centered care [4]. The Organization for Economic Co-operation and Development (OECD) has identified this complexity as an impending risk to the health of the Australian population that would require substantial revision of the Australian healthcare system to rectify [4]. Australians generally have good health outcomes, with at least the eighth highest life expectancy compared with other OECD countries [1] and a similar global position regarding burden of disease [5]. However, these positive health outcomes are not shared by Aboriginal people who are more likely to experience chronic diseases such as cardiovascular disease, diabetes, cancer and respiratory disease [6] and less likely to access preventive healthcare services [7].

Chronic diseases are associated with increased disability and poorer disease outcomes [6]. This increased disease burden contributes to Aboriginal Australians possessing an estimated life expectancy 10 years shorter than non-Aboriginal Australians [8]. The reasons for these inequities stem from the ongoing devastating impact of colonization on Aboriginal and Torres Strait Islander people that has resulted in trauma compounded by ongoing racism, discrimination, and loss of identity, language, culture and land all of which directly impact on healthcare outcomes [9]. Despite these inequities, Aboriginal and Torres Strait Islander peoples have survived and are resilient, maintaining connection to Country and culture.

Barriers to Aboriginal people accessing healthcare services are amplified by geographical factors. Aboriginal people are proportionally more likely to live in regional and remote areas, with 54% of Aboriginal Australians in NSW living outside of metropolitan areas [10]. Regional and remote areas receive less healthcare funding per capita, with very remote areas receiving less than a third of the funding of major cities [11]. People living in regional and remote areas are often required to travel to metropolitan centers to access services, particularly specialist services.

Aboriginal people are less likely to access mainstream health services and evidence suggests that when mainstream services are provided for Aboriginal people, they will have less positive health outcomes [7]. The customization of health services to culture and context is important to engage Aboriginal people into healthcare services and facilitate better outcomes [6]. Actions undertaken by the Australian government since colonization, including the removal of Aboriginal children from their families, have negatively impacted how safe Aboriginal people feel when accessing healthcare services [12]. Aboriginal people are less likely to access services until much later in the disease process and more likely to leave hospital early or not attend. This has led to the growth and success of ACCHOs since the 1970s [2,3].

Currently the Australian healthcare system lacks coordination across the various levels of Government and healthcare service providers [13]. This impacts on the provision of culturally safe, effective, and timely healthcare services for Aboriginal people. Improving the health of Aboriginal people requires a patient-focused, culturally safe approach at every level and component of the healthcare system, from national healthcare policy to local primary healthcare services [4]. Such an approach would require coordination at a national level as well as an appreciation of the interconnectedness of the entire healthcare system which may be achieved by adopting an ecological, or ecosystem perspective [14].

In order to understand the factors that influence access to healthcare services for Aboriginal people in regional and remote communities, this study applied an ecological standpoint informed by an ecological model [15] previously applied in a study with Aboriginal communities in Australia [16]. We aim to investigate the barriers and enablers experienced by Aboriginal people across the life course in accessing primary, specialist and allied healthcare services in regional and remote Australia, from the perspective of Aboriginal and non-Aboriginal healthcare workers who provide services in those communities.

## 2. Materials and Methods

Three communities in NSW were invited to participate in this qualitative study. These communities were identified as having proportionally high Aboriginal populations and a history of community-driven healthcare service development. They represented a variety of geographical characteristics being located on the NSW regional coast, in a remote area and a remote area adjacent to a state border. The authors had existing relationships with community health services within each community and the study utilized a co-design approach, including in the development of the interview questions and their face validity. Aboriginal members of each community, including representation from the local ACCHOs, collaborated to develop the research protocol for this study. The study reflects the concerns expressed by local Aboriginal people about the availability, connectedness and reliability of services and the priority to improve the match between healthcare services and community needs. The full protocol for this study has been described elsewhere [17].

Data were collected over a four-month period between October 2016 and January 2017. A snowballing sampling methodology was used to identify study participants who were involved with healthcare service delivery in each community. Services identified were ACCHOs and other government and non-government services, including not for profit organisations, that deliver primary, specialist and allied healthcare to Aboriginal people. Potential interviewees were approached by the researcher coordinating the study by telephone, face-to-face or by email, as appropriate, and invited to participate and/or suggest other suitable interviewees within the three communities. Interviews were conducted at a time and location convenient to the interviewees by one of two researchers, both of whom had training in qualitative research methods and prior experience of conducting interviews. Face-to-face interviews were conducted where feasible, or interviews occurred by telephone where distance, travel or time constraints existed. Interviews were audio-recorded and transcribed verbatim. Participants were asked a series of questions regarding healthcare service availability in their community. The questionnaire also included three open-ended questions about barriers and enablers towards Aboriginal people accessing healthcare services in their community service. These open-ended questions were: ‘In your community service how well do Aboriginal people access this service’; ‘In your community service, what do you think can be done to ensure Aboriginal people are given the opportunity to access healthcare services they need’; and ‘In your community service, what is not working in terms of Aboriginal people accessing healthcare services, and why do you think things are not working’. The interview duration was approximately between 15–90 min, with the average being 35 min.

Data were thematically analysed using inductive and deductive approaches [18]. Deductive thematic analysis was underpinned by an ecological framework [14]. One author conducted the thematic analyses and coding which was then reviewed and discussed by co-authors, including Aboriginal representation from each of the sites to reach consensus. Authors include Aboriginal community leaders and professionals who could ensure that findings reflected local community contexts and that their interpretation was valid. Ethics approval for this study was granted by the NSW Aboriginal Health & Medical Research Council (AH&MRC) Ethics Committee (1173/16).

## 3. Results

A total of 31 interviews were conducted in the three communities (*n* = 5 coastal, *n* = 13 remote, and *n* = 13 border). The professional roles of the interview participants included practice managers, Aboriginal Health Workers (AHWs), health service Chief Executive Officers, local nurses, Aboriginal Education Workers, school principals (of high school and primary school) and visiting cardiologist, ear nose and throat specialist, podiatrist and social worker. Two community members were also interviewed. Most participants identified as Aboriginal, but this was not recorded to preserve confidentiality of participants, particularly in small communities and remote locations.

Six themes were identified: (1) Improved coordination of healthcare services; (2) Better communication between healthcare services and patients; (3) Trust in the service provider and experience of cultural safety; (4) Importance of prioritizing health services towards key personal and community issues as defined by Aboriginal people; (5) Importance of reliable, affordable and sustainable healthcare services; (6) Distance and transport availability impact access to health care services. These themes were often present as both barriers and enablers to healthcare access for Aboriginal people, at all levels of the healthcare system and largely within each of the three communities.

Figure 1 presents an overview of the interplay of the themes across the ecological framework developed for this study, depicting the geographical context of the study in the three regional and remote communities, and findings through the local community lens. It highlights the relational nature of the different levels within the healthcare system; state-run health services are reflected in the funding, travel & accommodation subsidies as well as telehealth and the arrows show how these systems level aspects are experienced in the three communities across the six themes, at the local community level. For example, community perceptions of breeches of confidentiality at the service level were reported to impact individuals’ trust in that service and in turn can impact on service utilization reported through delayed presentations and inconsistent long-term care utilization for chronic conditions. Further, activity-based funding driven at the national level can impact the time clinicians spend with patients which in turn limits time spent on health education and time spent building a relationship with the patient. Some participants described the added complexities in health service provision and utilization for Aboriginal people given the intergenerational history of mainstream health services’ involvement in child removal, which participants reported continues to impact on issues of trust and safety which in turn impacts on delayed presentation to the service. Therefore, services providing culturally appropriate and relevant care, respectful and mindful of the added stress attending a health service can cause, are services participants reported to be more likely accessed by Aboriginal people. Finally, service utilization was reportedly increased when there was open and clear coordination and communication between services and across levels and where AHWs were integrated into healthcare delivery. An example of this was between the Local Health District and the community service to ensure specialist services were offered in accordance with community need.

### 3.1. Theme 1: Improved Coordination of Healthcare Services Is Needed

A lack of coordination of healthcare services was identified as a barrier to Aboriginal people accessing healthcare. Poor coordination led to inconsistency and under-servicing. This caused frustration among healthcare staff and Aboriginal people in regional and remote communities. One participant explained:
“I don’t know what we’re going to do about it, it drives us mad! It [service supply] has to be controlled from above us.”Remote community.

There was a perceived benefit of having increased interagency collaboration between healthcare services. This was best coordinated by a local community health representative and facilitated better access to healthcare services for Aboriginal people in the community. It was also a perceived barrier as it relied on a motivated individual to coordinate, the process of collaboration was unclear between agencies and some healthcare services were reluctant to share patient care. These factors impacted on the effectiveness of interagency collaboration.

“I think there needs to be regular interagency meetings and regular clinical handover meetings because we all work for different organizations and we all have different systems so if a patient came in tomorrow and saw me I would document it on [ACCHO] and then if the patient came in and saw Jo she would document it on [government health department] I feel the patient could be at risk from that and I think we should all be on the same page. I think we do it well but I think we could do it better. Handover is hard, can’t share knowledge.”Remote community.

Coordination was also very important at the level of the individual patient, as Aboriginal people often have complex care needs. The limited availability of healthcare services in regional and remote areas and the nature of these services require a high level of commitment and planning on the part of the patient to attend appointments. To support this, community healthcare workers, particularly AHWs, play a vital role in supporting one-on-one case management. This level of coordination required was also considered dependent on motivated healthcare workers to meet the health needs of Aboriginal people.

‘So, what—I rang Dental and I said to them, “What we need to do is you need to contact me. If any of those people identify as Aboriginal give me a call. I will then send my workers out, tap-tap on the door, ‘Hi. Do not forget you’ve got this appointment. Have you got transport? Do you need care for the kids?’”… just so the kids would actually get there.’Coastal community.

### 3.2. Theme 2: Better Communication between Healthcare Services and Patients

Poor communication between visiting healthcare services and communities was described. Local healthcare workers and patients were often unaware when specialist services were visiting, in some instances only finding out on the day of the visit. This was a barrier to Aboriginal people accessing visiting healthcare services but the role of AHWs in facilitating access was perceived as crucial:
“Unfortunately, without our health workers too many of the mob would miss out completely, a specialist would rock up and without the health workers he may sit there all day and see two people where the girls [staff] will go and door knock and get them.”Border community.

Within local communities, lack of communication between healthcare services was also a barrier to understanding what services are available. Like visiting services needing to notify the communities of their attendance; more communication is needed between local existing health services. One participant explained:
“Double up of services, for example counselling services because there’s someone at the AMS and someone here and we don’t communicate particularly to know which day they’re there and which day they’re here- and I think it would be really good if both those people would get together and say we’ll come then or whatever and even if they have the same patients which I think maybe this one comes and then goes over there to see someone else.”Remote community.

In addition to this, communication between healthcare services was considered critical to patient safety. Communication between agencies was considered an enabler to healthcare access and improving the quality of healthcare delivered. However, this depended on champions within the community to ensure interagency meetings took place.

“We have partnership with hospital working out pathways but we’re still working things out—we don’t get the discharge notification but through the partnership we now have the mental health teams with the AMS and at the hospital working together to work out pathways and a couple of staff at hospital pulled me aside and said you have to stay here because you’re getting things done.”Coastal community.

Communication between healthcare professionals and the patient was considered a barrier when medical advice was not communicated in a way that the patient could understand or when language used was not appropriate. It was apparent that this barrier could be overcome if there were trusted community members such as family or school staff present to provide further information, explanation or support:
“If anybody is confused at all in any way, shape or form they can say “Look can I see the educator?” And have an educator here full time.”Coastal community.

Furthermore, another example of the support from health workers to facilitate the communication between healthcare workers and patients was seen in a telehealth session with a paediatric specialist:
“We sit in with them so we know what’s being said because sometimes they use language that the parents don’t fully understand so then after the session they might turn around and go what did that mean and I then explain it to them what that actually meant.”Remote community.

### 3.3. Theme 3: Trust in the Service Provider and Experience of Cultural Safety

The ACCHO model was considered essential and effective in local service provision and led to the service being well utilized:
“Data shows we have a big increase in new patients; also shows people are coming back; and waiting room always full; people want access to services they need and they can get it a lot quicker than what they can get at the hospital; is a community environment; comfortable within their environment; people drive from the mission past the hospital to the AMS.”Coastal community.

Fear of racism, disrespect, judgement and negative government interventions were reported as barriers to Aboriginal people accessing some mainstream healthcare services. Fear of government involvement was evident. One participant said that there were lots of single parents with multiple children, and:
“If something happens to one child how do you fix that child while you are worrying about DOCS [Department of Child Services] and worrying about this, that and everything else and about being seen as a bad parent.”Coastal community.

Aboriginal health and education workers facilitate trust between healthcare workers and patients thereby enabling Aboriginal people better access to healthcare services. One study participant said that Aboriginal education workers can:
“Get that foot in the door, where then the nurses or health practitioners can then go in and those families don’t feel they’re being judged or they’re going to be reported to FACS [Family and Community Services] and that sort of thing.”Remote community.

The cultural competency of non-Aboriginal staff and services and developing trust and consistency was also considered important.

“I do agree that training Aboriginal staff is super important but also a mix can be good as well because they don’t always want to be seen by one of their mob, choice. Building trust is the most important thing and this takes a long time.”Border community.

### 3.4. Theme 4: Importance of Prioritizing Health Services towards Key Personal and Community Issues as Defined by Aboriginal People

Seeking healthcare was seen to compete unsuccessfully with ‘life issues’ more broadly. Healthcare workers can need to work closely with patients to encourage attendance at a service in this context, with one participant saying how patients:
“Access the service generally when they are sick. It is getting better to get them to come to a screening but it takes a lot of reminding and going back and forth. Other issues are I guess competing with life issues.”Border community.

However, the community-controlled model appeared to facilitate access for Aboriginal people:
“Community can see we must be getting something right, even before you can see change in attitude; I’m still new in town but people are coming up to me now, they’ll stop me at the cafe and they’ll say, what are you doing about …? Recently the breast screening came and the week before they said we’re following up from last year, they were going up to the hospital and then I said, hang on, last year I thought we agreed last year that you would come to the AMS—bring your van here … So they put the van at the AMS for the week and we filled every spot for that week even let white people come and use it. So it was a success that van being at the AMS.”Coastal community.

Having a choice in service provider was also considered important:
“I think there’s a little bit of that “it’s for white people” and the AMS is for Aboriginal…A lot of the time people aren’t looking for a service, they are asked by the staff if they would like to use it.”Remote community.

Aboriginal people may not access healthcare services due to the conflicting priorities of ‘sorry business’ and cultural practices in times of bereavement. One participant said:
“Sorry business really impacts on the community accessing services. I know a lot of clinicians get wild, not wild, they come out and are so community health oriented I suppose, if they’ve got appointments they think they should turn up and if they don’t turn up they think ‘oh, what’s going on?’ They’ve got to look at the broader picture around sorry business.”Border community.

Some study participants believed the barrier to service utilization was the healthcare services failing to meet the needs of the Aboriginal people within the community. Significant change was described as necessary to overcome this barrier.

“Until you get the process right in translating the state and commonwealth commitment to closing gap, until you get that right, they’ll be no results on the ground.”Coastal community.

### 3.5. Theme 5: Importance of Reliable, Affordable and Sustainable Healthcare Services

“These people deserve better, I think that this a beautiful community and I would like to see more resources for them, especially the elderly and the children, they deserve better.”Border community.

A key enabler to Aboriginal people accessing healthcare services in regional and remote communities was reported to be the availability of government subsidies. These included the Closing the Gap PBS Co-Payment and travel subsidies, which have been continuously implemented since 2010 [19]. The Closing the Gap PBS Co-Payment was described as having increased medication adherence for patients who would otherwise be unable to afford their medications:
“Since Closing the Gap (co-payment benefit) people are much more compliant with medications and I think it had been a fantastic thing.”Border community.

Government rebates paid to service providers for the provision of healthcare services through the Medicare Benefits Schedule was considered a barrier for Aboriginal people accessing reliable and consistent healthcare services across multiple sites. While the General Practitioner (GP) Management Plan aims to encourage GPs to provide comprehensive and individualized care, in some cases it had the opposite effect. In order to increase revenue, some healthcare services were reluctant to share the care of patients. One participant identified this as a reason for the lack of shared patient care between health services:
“Unfortunately that’s driven a lot of it by the service itself as well because they want the dollar to keep there. … They’re fighting over the GP plans.”Remote community.

### 3.6. Theme 6: Distance and Transport Availability Impact Access to Healthcare Services

Many healthcare services are provided to regional and remote areas by fly-in-fly-out (FIFO) services. Additionally, patients are often required to access healthcare services located in metropolitan centers. These factors were considered barriers to accessing healthcare services for Aboriginal people living in regional and remote communities. FIFO services in the communities that participated in this study were often provided by different healthcare workers treating patients for short periods of time. This was described as not sustainable because it provided limited opportunity for the development of relationships between patients and healthcare workers. In one community an enabler to Aboriginal people accessing healthcare services within the FIFO service model was having the same healthcare worker return to the community over an extended period. This continuity of care allowed healthcare workers to build rapport and establish relationships with patients and to provide culturally appropriate treatment and follow up:
“I have been here quite a while, the fact that I have been here and I am not going to come in from out of town and promise the world and never see them again. That is a big complaint from my clients.”Border community.

In the three communities that participated in this study, participants reported multiple transport services were provided within communities to facilitate access to healthcare services. While scheduled bus services were intended as an enabler to access healthcare services, the timing of transport services was described as an issue:
“They’re very limited in their funding. Community transport, now, again if you’ve got surgery going in the morning you’ve usually got to be there at six o’clock in the morning. They don’t open ‘til 9:00.”Coastal community.
“Transport is a big issue. Transport needs to be flexible and culturally appropriate.”Border community.

Healthcare service staff providing Aboriginal people with ad hoc transport to and from appointments enabled access to healthcare services but is impractical, unsustainable and time consuming for staff:
“Because we know it’s for our Elders, it’s out of respect and it’s about getting them a service, and we’re not a transport service but we do it. What? I’m gonna leave an Elder sitting there all frigging day who’s sick? No. That’s not on.”Coastal community.

## 4. Discussion

This study identified six themes present across multiple levels of the healthcare system as important to include in efforts to progress better access to healthcare services for Aboriginal people in Australia. Improving healthcare service delivery for Aboriginal people requires significant changes in the Australian healthcare system, but this is difficult given the system’s complexity and fragmentation. Applying an ecological approach [14] to this research highlights the interplay of attitudinal and structural factors across the system levels, particularly between the local community and individual levels. This is not unexpected given participants’ location within the diverse regional and remote communities. The richness of this largely community level data provides detailed information about experiences of the health care system, valuable insights into its strengths and weaknesses and factors to target in efforts to improve access to healthcare services for Aboriginal people.

Previous research has identified similar themes influencing access to healthcare for Aboriginal people but has often focused on barriers to accessing specific types of services or relating to specific conditions. Factors related to communication between healthcare services and patients, transport, experiences of cultural safety and cost of treatment have been identified as factors influencing Aboriginal peoples’ access to and utilization of cancer services [20], services following a cardiac event [21], and primary healthcare services [22]. The present study identified barriers and enablers to accessing both mainstream and Aboriginal community-controlled healthcare services, broadening the applicability of these findings.

Past enquiry has related predominately to barriers to Aboriginal people’s access to healthcare. While an understanding of barriers is important, it adds to the deficit discourse that characterizes much of the research in Aboriginal health [23]. In our study, we sought to identify service providers’ perspectives of factors that positively impact Aboriginal peoples’ experiences of, access to and utilization of healthcare. Several characteristics have been identified as evidence of the strengths-based approaches [23] that were apparent in the community settings of this study. These include resilience, cultural appropriateness, empowerment, and holistic approaches. We identified many Government and local community initiatives that support access to and enhance the experience and trust of Aboriginal people in health services. These include financial assistance, access to transportation options, interagency communication and coordination, continuity and consistency of care, one-on-one case management, respect for and understanding of factors competing with healthcare access. A further important contributor is provision of culturally respectful and regardful care, which was evidenced by the integral service provided by AHWs.

It is important to acknowledge the high burden faced by Aboriginal people in participating in death related rituals known as “sorry business”. Participation in sorry business is a frequent event in the lives of most Aboriginal Australians and typically takes priority over all other business including healthcare, employment and education. Participation in sorry business is not impeded by age or distance. Understanding the frequency of sorry business and accommodating this within service design and practice will likely increase the cultural safety of services and participation in those services by Aboriginal people [24].

Knowledge of the barriers and enablers to accessing healthcare for Aboriginal people is important in understanding how to improve the provision of healthcare services for Aboriginal people living in regional and remote areas. However, system-wide change is required to achieve this. Further research should explore new programs and policy solutions to overcoming these barriers identified in accessing healthcare for Aboriginal people, building on the enablers identified in this study. In addition, there should be continued support for these enablers to healthcare access. These included transport and pharmaceutical subsidies, the latter which have been found to increase medication use nationally since the reduction or elimination of medication co-payments for Aboriginal Australians with chronic disease [19]. Our findings also point to greater support for community health personnel who facilitate communication between Aboriginal people and health service providers, particularly the key role of AHWs and expansion of these roles [25,26].

The inconsistency in healthcare workers attending the communities through FIFO services was identified as a barrier to healthcare access for Aboriginal people because it prevented development of relationships between healthcare workers and their patients. Despite this, FIFO is currently considered beneficial in the short-term as a way to deliver high quality specialist services to the small, widely dispersed [25] communities in Australia [27]. We suggest, from these findings, several strategies to try and overcome barriers inherent in FIFO services. These are: having consistent service providers to build rapport and relationships with patients; providing communities with adequate prior notification of attendance and incorporating the use of telehealth for follow up with the aid of local healthcare workers to facilitate communication. This is consistent with a recent review that recommends FIFO services should only be utilized in combination with the development of well-resourced and staffed primary healthcare services in regional and remote communities [27].

Some healthcare services, such as those requiring specialist facilities, require patients to travel to metropolitan centers and regional hubs to receive care. Previous research suggests that for Aboriginal people to travel to access healthcare means being separated from Country and family and support networks and can impact on all elements of the multidimensional concept of health and wellbeing as described by Milroy’s Dance of Life [28]. Further, while government subsidies are available to pay for travel and accommodation costs incurred by Aboriginal people when travelling to access healthcare, people may be required to pay up front costs, and not all associated costs may be covered [29]. Therefore, there may be an additional, structural financial barrier to Aboriginal people living in regional and remote areas in accessing specialist and complex healthcare services in metropolitan centers.

Our findings highlighted the integral role and importance of AHWs in health service delivery. AHWs initiated interagency communication, facilitated communication, were often the conduit between other healthcare workers and patients, transported patients to and from appointments and provided one-on-one case management. This is consistent with previous research that reported ACCHOs are already implementing many important workforce development strategies that are having a positive impact on health service utilization and patient outcomes [30]. Our findings highlight that these approaches should be tailored to local needs and contexts. To improve access to healthcare services for Aboriginal people, such roles should be formalized, recognized and funded appropriately. Health system navigation, led by AHWs or trained community members, and its potential to facilitate access could also be explored.

Consistent with previous research, this study shows the impact of colonization and the removal of Aboriginal children from their families both historically and currently resulting in Aboriginal people being afraid to access healthcare services. Building trust among Aboriginal people and communities is important to overcome this barrier [31]. It has been previously documented that mainstream services lack cultural competence and acceptability among Aboriginal people [32]. Some strategies to create culturally competent healthcare services include partnership with Aboriginal communities and Elders [33], offering flexibility of services, working in partnership with families, supporting coordination of healthcare, prioritizing employment and retention of Aboriginal staff. Providing staff training in cultural competence [12] and encouraging reflexive and regardful care [34] are also important.

Strengths of the study include the codesign process with the participating communities and the use of an ecological or ecosystem approach [35] to contextualize the findings. A limitation of this study is that it was conducted in just three regional and remote communities in NSW. These findings represent the barriers and enablers experienced by Aboriginal people living in those communities, that we have noted have a particular history of community driven healthcare service development and may experience fewer barriers than other communities. Therefore, further research is warranted to explore if these findings are applicable to Aboriginal people across urban and other regional and remote areas in NSW and Australia. In addition, the barriers and enablers identified in this study were primarily from the perspective of Aboriginal and non-Aboriginal healthcare workers and people involved in healthcare delivery. Further research to explore whether these findings are consistent with barriers and enablers as perceived by Aboriginal people living in the three communities is important.

## 5. Conclusions

There is an interplay of factors across the health system that impact health service access and utilization for Aboriginal people, as evident in the six themes that were identified in this study within three communities. Enabling factors were coordination of healthcare services within jurisdictions, effective communication between healthcare services, trust in health services and positive experiences of cultural safety, prioritization of access for Aboriginal people, resourcing for healthcare services and addressing distance and transport barriers. Identifying these barriers and enablers provides an opportunity to improve access to healthcare services for Aboriginal people within Australia’s complex and fragmented healthcare system, while maintaining a patient focus. This provides strategic options to improve healthcare service accessibility in a complex system and with consideration of geographical and local contextual factors.

## Figures and Tables

**Figure 1 ijerph-18-03014-f001:**
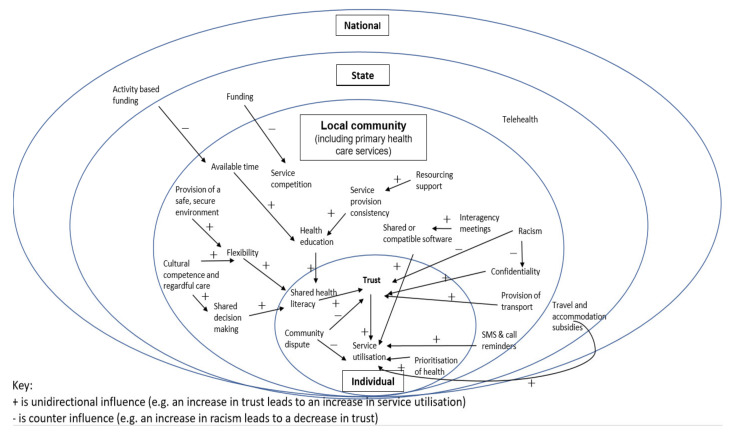
Interplay of themes across the ecological framework.

## Data Availability

Data are available upon request to the authors and conditional on ethics approval.

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
