# Peer review of "Enablers and Barriers to Accessing Healthcare Services for Aboriginal People in New South Wales, Australia"

_ijerph, 2021, doi:10.3390/ijerph18063014_

Round 1

Reviewer 1 Report

Thank you for the opportunity to review ‘Enablers and barriers to accessing healthcare services for Aboriginal people in New South Wales, Australia’.

This is valuable and important study exploring professional’s views about access to the worker’s healthcare service for Aboriginal people living in rural and remote NSW.

Title

The title needs to be type set to avoid the use of a hyphen.

Abstract

There seems to be an inconsistency in the aim ‘Aboriginal people living in rural and remote Australia’ and the way the scope is reported in results: ‘Thirty-one interviews were conducted (n =5 coastal, n=13 remote, and n=13 border)’. It would be good to be clearer about the breakdown in terms of  remote and rural.

Introduction

The study aim refers to Aboriginal people. Does this include people across the age span?

Methods

The paper reports the professional’s role only, not service type. The interview questions focus on the worker’s service. Therefore, it would be good to know the types of services that were included in the study to understand the scope of services included.

Results

‘Figure 1 presents an overview of the interplay of the themes across the ecological framework developed for this study. It highlights the relational nature of the different levels within the healthcare system’. Figure 1 is divided up using national state local and individual sections. However, this figure seems to relate to geographical context, not the structure of the health system. Are state-run health services and federally-funded GPs included in this figure? They seem to be missing. Again, it is important to understand the scope of the services included in the study.

Theme 5 has three elements: ‘Importance of reliable, affordable and sustainable healthcare services’.  However, the examples seem to focus more on affordability. It would be good to clearly illustrate all three elements of the theme in the examples provided.

Discussion

The discussion includes comments about the types of services that should be strengthened. This would be stronger if we knew the types of services included in the study, and therefore, the study scope.

The focus of this study is on access. I wonder if future research could focus on the related but broader concept of health system navigation?

I wonder if you considered the literature about types of barriers, for example: structural, attitudinal, and relating to age?

Author Response

Thank you for your comments and for the opportunity to revise this paper. Please find our point-by-point responses to the reviewer comments for your further consideration.

Comment

Author response

Reviewer 1

Thank you for the opportunity to review ‘Enablers and barriers to accessing healthcare services for Aboriginal people in New South Wales, Australia’.

This is valuable and important study exploring professional’s views about access to the worker’s healthcare service for Aboriginal people living in rural and remote NSW.

Title

The title needs to be type set to avoid the use of a hyphen.

 Thank you for your helpful review and suggestions. We have amended the title to avoid the hyphen and this can be ensured in the journal editing should the paper be accepted.

Abstract

There seems to be an inconsistency in the aim ‘Aboriginal people living in rural and remote Australia’ and the way the scope is reported in results: ‘Thirty-one interviews were conducted (n =5 coastal, n=13 remote, and n=13 border)’. It would be good to be clearer about the breakdown in terms of remote and rural.

We have clarified in the Abstract methods that the three communities – coastal, rural, and remote - represent three different and diverse regional and remote locations. These three communities fit within the Australian Institute of Health and Welfare regional and remote definitions that is referenced in the Introduction (reference 11).

Introduction

The study aim refers to Aboriginal people. Does this include people across the age span?

We have clarified that the study pertains to people across the life course (line 116).

Methods

The paper reports the professional’s role only, not service type. The interview questions focus on the worker’s service. Therefore, it would be good to know the types of services that were included in the study to understand the scope of services included.

We agree that information on service type is important to include and have added that the studies recruited staff from Aboriginal community-controlled health organisations (ACCHOs) and other government and non-government services that deliver healthcare to Aboriginal people (lines 133-135) and clarified, consistent with the study protocol, that the study considered primary, specialist and allied healthcare services.

Results

‘Figure 1 presents an overview of the interplay of the themes across the ecological framework developed for this study. It highlights the relational nature of the different levels within the healthcare system’. Figure 1 is divided up using national state local and individual sections. However, this figure seems to relate to geographical context, not the structure of the health system. Are state-run health services and federally-funded GPs included in this figure? They seem to be missing. Again, it is important to understand the scope of the services included in the study.

Theme 5 has three elements: ‘Importance of reliable, affordable and sustainable healthcare services’.  However, the examples seem to focus more on affordability. It would be good to clearly illustrate all three elements of the theme in the examples provided.

The Figure does depict the geographical context of the study, being three regional and remote communities, and findings through the local community lens. State-run health services are reflected in the funding, travel & accommodation subsidies, and telehealth and the arrows show how these systems level aspects are experienced in the three communities across the six themes, at the local community level. We have explained this in lines 177 – 182.

We have clarified how these examples also relate to reliability and sustainability. We consider that the Closing the Gap co-payment to be an example of a sustainable subsidy as it has been continuously implemented for over a decade, since 2010 and have clarified this in lines 349-50. We have also clarified that “Government rebates paid to service providers for the provision of healthcare services through the Medicare Benefits Schedule was considered a barrier for Aboriginal people accessing reliable and consistent healthcare services across multiple sites” (line 357)

Discussion

The discussion includes comments about the types of services that should be strengthened. This would be stronger if we knew the types of services included in the study, and therefore, the study scope.

The focus of this study is on access. I wonder if future research could focus on the related but broader concept of health system navigation?

I wonder if you considered the literature about types of barriers, for example: structural, attitudinal, and relating to age?

We trust that the types of services, ACCHOs and other and other government and non-government services that deliver primary, specialist and allied healthcare, is now clearer in the Methods and have also clarified in the Discussion that the service scope includes community-controlled, government and non-government services.

Health system navigation is certainly relevant though the focus of the present study was on access of provision. We have made this clearer in lines 488-490 and highlighted how improved access could facilitate health system navigation and recommend it as a potential avenue for investigation.

We have clarified, on your useful suggestion, that the study scope considers Aboriginal people across the life course and this is reflected through the findings and themes, so we do not consider age-related barriers to be relevant enough to warrant further discussion. Literature categorising barriers as attitudinal and structural primary relates to mental health which we do not feel would be appropriate and relevant to our broader study, nor take account of but we have added clarification where barriers reported in our study may be considered structural or attitudinal (lines 404; 477)

Reviewer 2 Report

Thank you for the opportunity to review this manuscript which explores barriers and enablers to Aboriginal people accessing healthcare in NSW. It is an interesting and well written article with strong co-design with the Aboriginal communities involved. I recommend publication of the article, and have only a few minor comments for the authors to consider.

Introduction

There is use of both acronyms: ACCHO’s versus ACCHSs?

Line 72 is missing a full stop.

Methods/discussion

I wonder if a couple of features of the sample could be highlighted or discussed as potential limitations to generalisability of the findings. These are:

  1. that the 3 communities selected “had a history of community driven healthcare service development”. This may mean that findings may not apply (and there are likely to be additional barriers?) in communities without the same history; and
  2. The mix of people who were interviewed- mainly providers. While 2 community members were included, community members may have a different perspective to providers and others about barriers/enablers to accessing healthcare?

Author Response

Thank you for your comments and for the opportunity to revise this paper. Please find our point-by-point responses to the reviewer comments for your further consideration.

Comment

Author response

Reviewer 2

Thank you for the opportunity to review this manuscript which explores barriers and enablers to Aboriginal people accessing healthcare in NSW. It is an interesting and well written article with strong co-design with the Aboriginal communities involved. I recommend publication of the article, and have only a few minor comments for the authors to consider.

Introduction

There is use of both acronyms: ACCHO’s versus ACCHSs?

Line 72 is missing a full stop.

Thank you for your review and supportive comments. We have now used ACCHOs consistently and have added the full stop.

Methods/discussion

I wonder if a couple of features of the sample could be highlighted or discussed as potential limitations to generalisability of the findings. These are:

that the 3 communities selected “had a history of community driven healthcare service development”. This may mean that findings may not apply (and there are likely to be additional barriers?) in communities without the same history; and

The mix of people who were interviewed- mainly providers. While 2 community members were included, community members may have a different perspective to providers and others about barriers/enablers to accessing healthcare?

We agree that these are important limitations to consider and have added our reflections on lines 507-509 and 515.

Small changes are suggested in post-it notes in draft uploaded.

Thank you for your review, we have incorporated your suggestions in the revised version.

Reviewer 3 Report

Small changes are suggested in post-it notes in draft uploaded. 

Author Response

Thank you for your review, we have incorporated your suggestions in the revised version.